# Area extraction and spatiotemporal characteristics of winter wheat–summer maize in Shandong Province using NDVI time series

**Chao Dong[1,2], Gengxing Zhao[2]\*, Yuanwei Qin[3], Hong Wan[1]**

**1** College of Information Science and Engineering, Shandong Agricultural University, Tai'an, Shan Dong, China, **2** College of Resources and Environment, Shandong Agricultural University, Tai'an, Shan Dong, China, **3** Center for Spatial Analysis, College of Atmospheric and Geographic Sciences, University of Oklahoma, Norman, Oklahoma, United States of America

\* zhaogx@sdau.edu.cn

**Data Availability Statement:** All relevant data are within the manuscript and its Supporting Information files.

## Abstract

The use of remote sensing to rapidly and accurately obtain information on the spatiotemporal distribution of large-scale wheat and maize acreage is of great significance for improving the level of food production management and ensuring food security. We constructed a MODIS-NDVI time series dataset, combined linear interpolation and the Harmonic Analysis of Time Series algorithm to smooth the time series data curve, and classified the data with random forest algorithms. The results show that winter wheat–summer maize planting areas were mainly distributed in the western plains, southern region, and north-eastern part of the middle mountainous regions while the eastern hilly regions were less distributed and scattered. The winter wheat–summer maize planting areas in the study area continued to grow from 2004–2016, with the most significant growth in the northern part of the western plains and Yellow River Delta. The spatial planting probability reflected the planting core area and showed an intensive planting pattern. During the study period, the peak value and time for the NDVI of the winter wheat were significantly different and showed an increasing trend, while these parameters for the summer maize were relatively stable with little change. Therefore, we mapped a spatial distribution of the winter wheat and summer maize, using the time series data pre-processing synthesis and phenology curve random forest classification methods. Through precision analysis, we obtained satisfactory results, which provided a straightforward and efficient method to monitor the winter wheat and summer maize.

## Introduction

Wheat and maize are the main food crops in northern China and key components in food production [1, 2]. Satellite remote sensing technology can rapidly and accurately obtain information on crop planting areas, monitor planting status, and simultaneously map spatial distribution patterns, influencing factors, and planting probability while estimating crop

**Funding:** This work was supported by the National Key Technology Research and Development Program of the Ministry of Science and Technology of China (2015BAD23B0202), the National Natural Science Foundation of China (no. 41877003), the fund award for Double-First Class (SYL2017XTTD02), and the Major scientific and technological innovation projects in Shandong Province (2019jzzy010724).

**Competing interests:** The authors have declared that no competing interests exist.

output, water, fertilizer demand, and other information [3–6]. These data can provide important reference information for crop production planning management, making remote sensing monitoring of wheat and maize an important means to ensure food security by improving production management.

In crop remote sensing monitoring, previous studies have investigated area mapping and change monitoring of crops using higher resolution remote sensing images, such as the landsat and SPOT images, to map winter groundcover on agricultural fields [7]. RapidEye images were used to derive training data for the characterization of time-series Landsat data [8]. High spatial resolution remote sensing data can reduce mixed (boundary) pixels and improve classification and recognition effects and accuracies [9] but performing out real-time and effective long-term monitoring of large areas due to small coverage areas, long return-visit periods, low spectral resolutions, and other effects is difficult. Low-resolution satellite sensors, such as the Moderate Resolution Imaging Spectroradiometer (MODIS), have 36 bands, higher temporal resolutions, and longer service cycles, which are suitable for monitoring continuous changes in crops at large scales and over long time periods [10, 11].

Crop phenology is an appropriate growth and development rhythm formed by an organism's long-term adaptation to temperature conditions. Crop phenological information is not only the basis for agricultural production decisions, but also an important parameter for crop simulation models [12]. Understanding phenological changes in crop monitoring and ecological prediction is highly important. To solve problems associated with low spatial resolution in MODIS satellite data that affect the classification accuracy, phenological metrics can be introduced to improve the classification accuracy [13]. Descriptive data on crop growth provided by shorter return-visit satellites are combined into time series datasets to reflect crop phenological characteristics. Phenological characteristics can provide information other than spatial spectral characteristics. As different vegetation has unique phenological characteristics, phenological characteristics are considered a remote sensing method whose application can improve the accuracy of the vegetation classification [14]. Wardlow and Egbert [15] discussed the curve differences in the growth period of six crop types in the Central Plains of the United States based on NDVI time series data and phenological metrics of different crops, indicating that the NDVI values of different crop types can be classified according to the time series of the different crop types. Based on NDVI time series data and the characteristics of two seasonal crops, Li et al. [16] extracted crops, discussed the spatiotemporal patterns of the crop phenological period/ planting system in north China, and verified that the quantity distribution and spatial distribution of crop phenological characteristics were significantly different in the different growing seasons. Based on these phenological differences, remote sensing extraction can resolve the foreign body homology problem and improve the classification accuracy. For how to use remote sensing data to indicate phenological information, the current research is mainly focused on single satellite data, but less on double satellite data. In addition, for example, to improve the quality of data set and to better characterize the phenological curve need to be studied.

To improve the accuracy, previous studies have performed a significant amount of research on the production and processing of time series datasets. Gao et al. [17] produced the Landsat-MODIS time series dataset, which fused two types of satellite data and mapped crop growth. Zhou et al. [18] compared the remote sensing time series reconstruction models at different time intervals, while Liang et al. [19] compared the reconstructed time series data via linear interpolation and with a smoothing algorithm, which improved the quality of the NDVI time series dataset. Previous studies have shown that both the Harmonic Analysis of Time Series (HANTS) and Savitzky–Golay methods are powerful tools to reproduce NDVI time series data [20, 21]. In the specific classification method, previous studies have also made various attempts to improve the characteristics of NDVI time series data curves. Atzberger and Rembold [22]

used a neural network method to map the spatial distribution of winter crops in central Italy. Zhu et al. [23] used the spectral angle clustering partition method to estimate the winter wheat area from MODIS time series data. Manfron et al. [24] proposed a rule-based analysis method to estimate the interannual sowing date of winter wheat using MODIS time series data. The application of these studies and methods has achieved good results.

In general, current research on remote sensing monitoring of crops based on phenological metrics focuses on the construction of high-quality timing datasets, the selective and precise evaluation of classification models, and the analysis of extraction results. Time series data can reflect crop phenological characteristics well, but their spatial resolution is generally low, as well as more mixed pixels. Due to the large time span and noise in the data, we must improve our method of constructing high-quality time series datasets. In addition, the classification of time series data differs from traditional data classification. Therefore, we must further explore more efficient and accurate classification methods and fully exploit the advantages of time series data classification.

This study uses Terra and Aqua satellite data to construct a MODIS-NDVI time series dataset, using quality control data and the HANTS method to improve the quality of the time series data, as well as using the random forest method to extract information about the distribution of winter wheat–summer maize planting in Shandong Province from 2004 to 2016. Thereafter, we analyzed the spatiotemporal pattern of wheat planting and its changing motives, revealing its spatiotemporal patterns, influencing factors, providing basic data, and a scientific basis for the planting and management of winter wheat in Shandong Province and northern China. The objective of this study is to be able to quickly draw crop maps with high precision. This method makes full use of the advantages of time series data to reflect the phenological characteristics. Combined with the wide coverage of MODIS data, this method can better achieve this objective.

## Data and methods

### Study area

Shandong Province is located in the North China Plain between the lower reaches of the Yellow River and eastern coast (Fig 1). Shandong Province consists of two parts, i.e., the peninsula and inland region, with a land and cultivated areas of 15.71 of 7.52 Mha, respectively. The study area can be divided into three landforms: hilly areas of the middle southern region, western plains, and eastern hilly regions. The province has an average of 2,290–2,890 hours of sunshine per year, such that the heating conditions can conform to the needs of two crop rotations per year. There are currently both double- and single-cropping systems in Shandong Province [25], where the main grain crops are wheat and maize.

### Satellite data

This study used MODIS-NDVI products to construct the time series datasets. The vegetation index product uses quality-driven, angle-constrained, and maximum-value composites to ensure data quality. Additionally, through the bidirectional reflectance distribution function [26, 27] model correction, the observation value of the sensor's viewing angle is unified as the observation point of the sub-satellite point. NDVI is the best indicator of vegetation growth status and vegetation coverage information, which can be calculated with the following expression:

$$\text{NDVI} = \frac{(\text{NIR}) - (\text{R})}{(\text{NIR}) + (\text{R})} \tag{1}$$

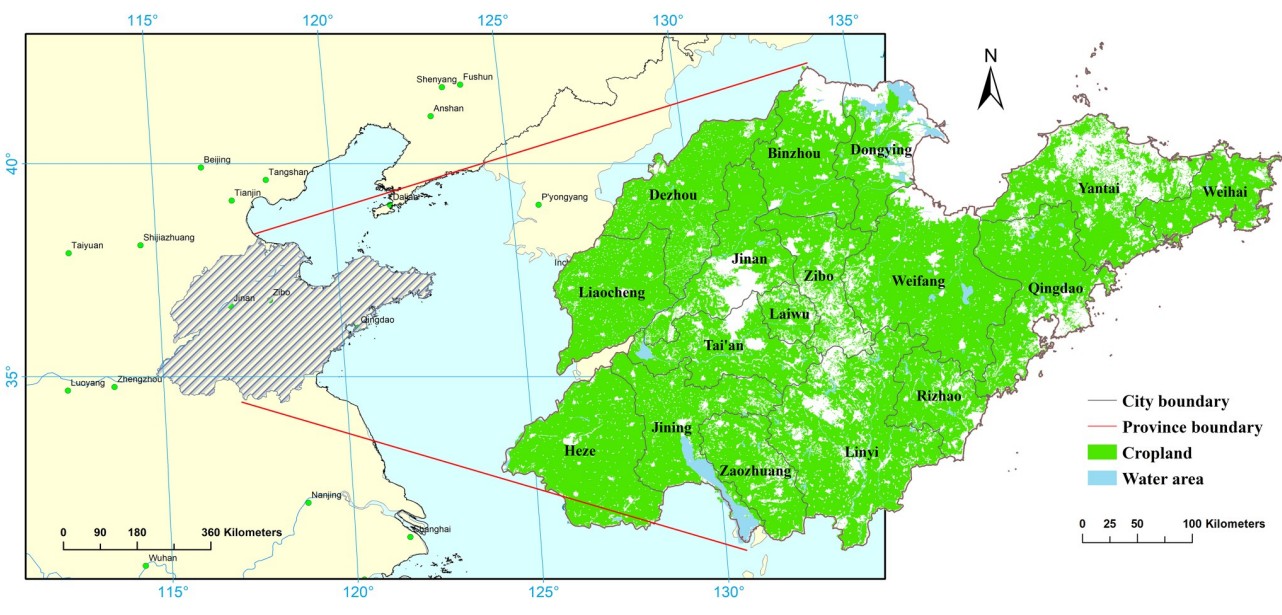

**Fig 1. Study area and cropland distribution.**

where NIR is the reflectance value in the near-infrared (0.84–0.88 μm) range and R is the reflectance value of the red band (0.62–0.67 μm). The NDVI value is normalized to between –1 and 1, where positive values indicate vegetation coverage and increase with increasing coverage [28].

The satellite data used were obtained from Level 1 products and the Atmosphere Archive and Distribution System (https://ladsweb.modaps.eosdis.nasa.gov). We used the vegetation indices of 16-day composites from Terra/MODIS (MOD13Q1, h27v05) [29] and Aqua/MODIS (MYD13Q1, h27v05) [30] at a spatial resolution of 250 m. Each satellite contains 23 phase nodes each year, such that the time series has a total of 46 scene data points. The study period spans from 2004 to 2016, with a total of 598 scene images.

## MODIS time series pre-processing

**NDVI time series dataset construction.** First, we extracted the NDVI index data in MYD13Q1 and MOD13Q1 using the MODIS Reprojection Tool, which were projected into a Universal Transverse Mercator projection. As the double satellite adopts the 'half-synthetic period dislocation' method, each time node is different by eight days, such that we sorted the NDVI data by the annual day-of-year time and combined it into a MODIS-NDVI time series dataset. The time series covers the complete growth period of the ground crops (Fig 2). The composite data can reduce the influence that both clouds and snow have on the data, generating more detailed timing curves to facilitate the extraction of timing features. We then applied the Shandong boundary mask to the time series data.

**Cloud and snow pixel value processing.** MODIS quality control data (QC) include both pixel reliability and vegetation index quality. The former uses binary numbers to indicate the quality of the image's metadata, such as high-quality data (i.e., a value tagged with 0), marginal data (i.e., a value tagged with 1), and cloud and snow data [31]. We developed the program to process the MODIS-NDVI time series data using the Interactive Data Language. For pixels labelled as clouds and snow, the original value was replaced by linearly interpolating the data

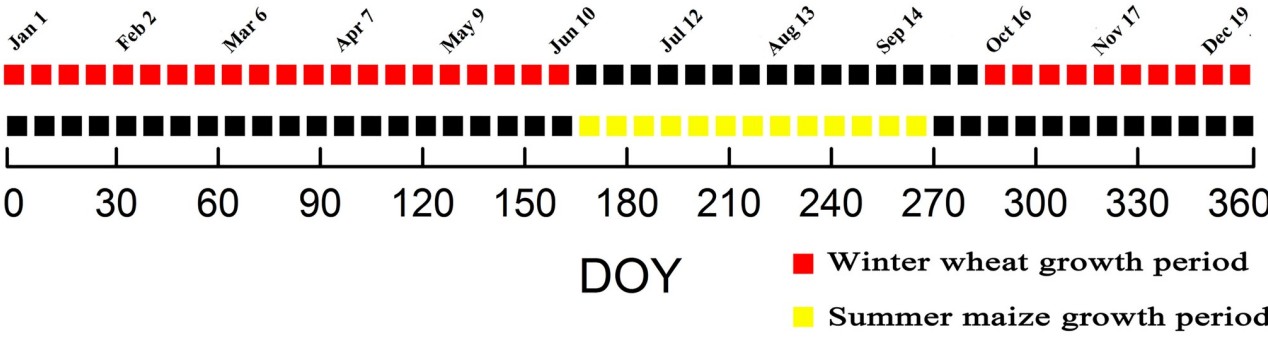

**Fig 2. Node diagram for the time series data.** (DOY: day of year).

nodes labelled as high-quality with the adjacent time series data, retaining the corresponding marks for high quality and marginal data. This can eliminate, as much as possible, the influence of outliers caused by clouds and snow in the time series data and improve data quality (Fig 3).

**Time series smooth processing.** To further remove noise and maintain data consistency, the MODIS-NDVI time series data were smoothed by HANTS to improve the quality of the time series data [32, 33]. The HANTS method is a phenological analysis method that time-interpolates time series values by a Fourier transform and least squares method to obtain a new, smoother time series. The four main control parameters in the HANTS module operation include valid data min and max, number of frequencies, fit error tolerance, and degree of over determinedness. In this study, we set the four parameters to –2,000 to 9,000, 3, 1,000, and 8, respectively, according to the test. We used such a high permissible valid data min and max because the value of the MODIS NDVI product was multiplied by 10,000. After smoothing, the influence of noise was removed from the data, such that it better reflects the internal laws of the crop growth curve.

## Mapping winter wheat and summer maize

Based on the MODIS-NDVI time series dataset and phenological characteristics, we extracted the winter wheat–summer maize planting areas. The extraction steps were as follows. 1) We established a time series interpretation mark for the study area. Land cover in the study area was divided into five categories, i.e., winter wheat–summer maize, one-season crop area, built-up area, forest and grassland area, and water area. Based on these categories, the category sampling points were generated. The data category labels were further determined based on the high-resolution satellite images from Google Earth. The sampling point used the spatial analysis to extract the timing curve from the MODIS time series. We calculated the mean and standard deviation of the curve by category and plotted it. The temporal curve shows the phenology of the vegetation above this category. We used the curve as a sign of interpretation. 2) We established a classification training dataset. Considering the significant differences between the east and west in Shandong Province (a longitudinal difference of 8˚), as well as regional differences in the growth environment, the study area was divided into 50,000 m × 50,000 m grids, within which training points were allocated according to the complexity of the land cover type. Attribute labelling of the training points was mainly based on feature discrimination of the time series curve. Pixels with mixed-ground objects, were assisted by high-resolution satellite images. 3) We completed random forest classification [34]. Random forest classification is a combination classification model composed of numerous decision-tree

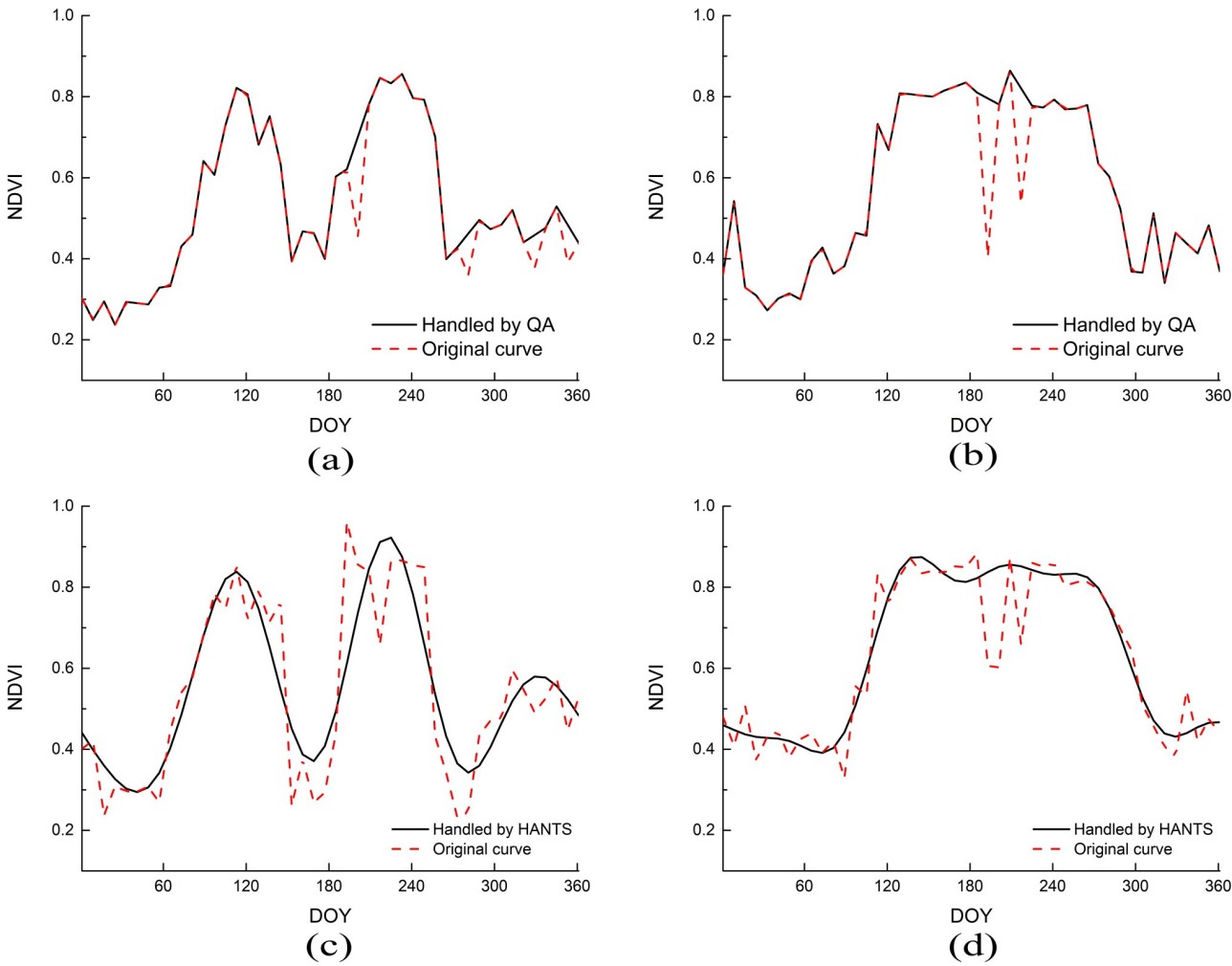

**Fig 3. A comparison of the temporal data processing.** Note: (a) and (b) are handled by QA comparisons with the raw data while (c) and (d) are handled by HANTS comparisons with raw data.

classification models. This model has a good anti-interference ability for over-fitting and training data outliers. The first RF Classification (RFC) model was parameterized using training points (i.e., the phenological features of ground objects) for training and internal validation. The value for the number of trees used was 100. The function to determine the number of randomly selected features was the square root of the number of all features. In a second step, the phenology model was used to perform the MODIS time series. This study used the random forest module provided by ENMAP Box V2.2 to classify the time series data of the study area [35].

## Accuracy assessment

To evaluate the comprehensiveness and objectivity of the winter wheat–summer maize extraction results, the accuracy evaluation was divided into classification and mapping accuracy and area accuracy. We generated validation datasets and used statistical data from the planting area to evaluate the respective accuracy of the methods. The validation dataset was based on the land use status data of the study area and high spatial resolution images from Google

Earth. The area accuracy assessment data were obtained from the statistical data on the wheat and maize sown area in the Shandong Statistical Yearbook [36] from 2004 to 2016. The data were evaluated with a regression analysis of the remotely sensed wheat and maize area. To further evaluate the area extraction accuracy, the consistency index of the extraction results were calculated with the following expression:

$$c = \left(1 - \frac{|x - y|}{y}\right) \times 100 \tag{2}$$

where $x$ is the remote sensing extraction area, $y$ is the statistical area from the Shandong Statistical Yearbook, and $c$ is the consistency index. We then compared the differences in the consistency indices between the study area and each city to determine the parameters that influenced the extraction accuracy.

### Spatiotemporal analysis

Statistical results of the area extraction at different time phases were used to analyze the interannual variation in the winter wheat–summer maize area. The wheat distribution maps from 13 years were compared, analyzing their spatial distribution patterns and dynamic characteristics. Based on the distribution map, the planting probability can be calculated by counting the amount of wheat and maize during the statistical time period. The probability map of the winter wheat–summer maize planting from 2004 to 2016 allowed us to analyze the spatial stability of the planting area. Furthermore, by comparing the peak differences in the NDVI time series curves in each year, we analyzed the dynamic changes in the status of winter wheat and summer maize growth.

## Results

### NDVI time series data profile

The solid black line in Fig 4 is the mean value of the NDVI time series curve at the sampling points of winter wheat and summer maize. This line shows that the NDVI time series curve of the winter wheat–summer maize area has a typical bimodal curve. The first peak occurs between days 60 and 170 during the winter wheat growth cycle while the second peak appears between days 180 and 270 during the summer maize growth cycle, which is consistent with the growth cycle of winter wheat–summer maize.

The dotted line in Fig 4 represents the standard deviation of the NDVI values for the nodes of the winter-summer maize, which reflects the uncertainty in the variation of the local classes at the time nodes. The annual standard deviation of winter wheat and summer maize is characterized by a low state, which indicates that the curve characteristics are stable.

### Precision validation

**Accuracy assessment.** Table 1 lists the statistical results of the classification accuracy from 2016. The overall accuracy was 90.75%, with a Kappa coefficient of 0.88. Among these values, the classification and mapping accuracy of water areas was the highest at 96.43%. The user's accuracy for the winter wheat–summer maize was 95.80% while the producer's accuracy was 88.37%. The classification accuracy of the winter wheat–summer maize was higher in all land categories. The classification results show that the random forest classifier had higher classification accuracy for the representations by time series features, such that clearer time series curve features resulted in increased classification accuracy.

 

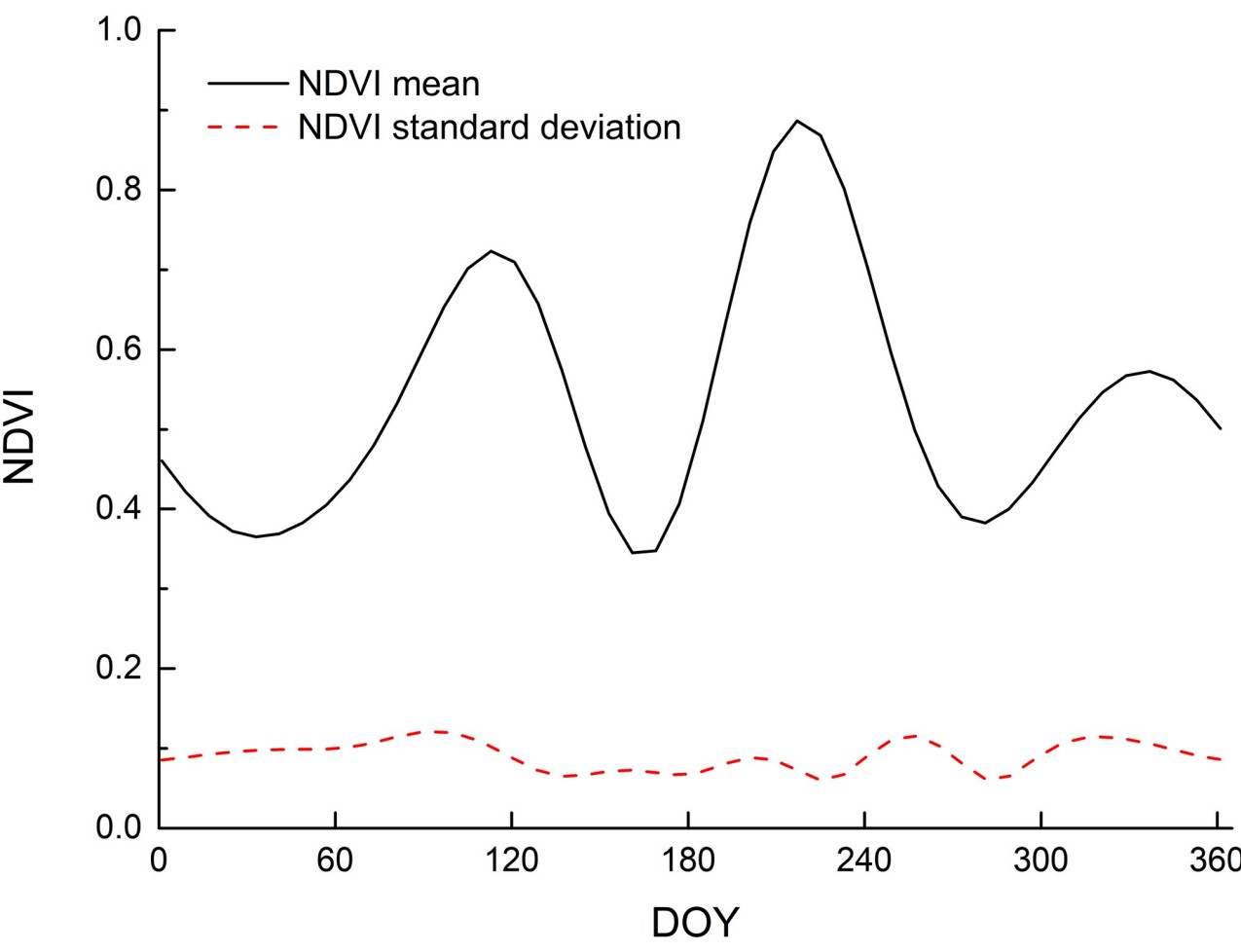

**Fig 4. NDVI temporal curves of the winter–summer maize.** (DOY: day of year).

Table 2 lists the classification accuracy of the winter wheat–summer maize from 2004 to 2016. The results show that the user's accuracy for winter wheat–summer maize land ranged from 91.8 (2014) to 96.4% (2006) while the producer's accuracy ranged from 84.1 (2007) to 93.3% (2015). The 13-year average user's accuracy was 94.5%, while the average producer's accuracy was 90.2%. Overall, the distribution of the classification accuracy for the winter wheat–summer maize in each year was relatively balanced, resulting in higher average classification accuracy.

**Table 1. Classification accuracy in 2016.**

| Map class | User's accuracy (%) | Producer's accuracy (%) |
|---|---|---|
| Winter wheat–summer maize | 95.80 | 88.37 |
| Forest and grass | 86.84 | 89.19 |
| Water | 96.43 | 96.43 |
| Built-up land | 92.31 | 93.75 |
| One-season crop | 80.82 | 89.39 |
| Overall accuracy 90.75% | Kappa accuracy 0.88 | |

**Table 2. Accuracy rating table.**

| Year | 2004 | 2005 | 2006 | 2007 | 2008 | 2009 | 2010 | 2011 | 2012 | 2013 | 2014 | 2015 | 2016 |
|---|---|---|---|---|---|---|---|---|---|---|---|---|---|
| User's accuracy (%) | 94.7 | 93.8 | 96.4 | 93.9 | 93.2 | 93.7 | 94.1 | 93.2 | 96.1 | 95.7 | 91.8 | 95.9 | 95.8 |
| Producer's accuracy (%) | 86.1 | 92.1 | 92.9 | 84.1 | 92.8 | 91.6 | 91.3 | 92.2 | 91.7 | 92.2 | 84.4 | 93.3 | 88.4 |

**Accuracy verification based on area statistics.** Fig 5 shows a scatter plot of the remote sensing extracted and statistical areas from the Yearbook for Shandong Province and other cities from 2004 to 2016. Based on the provincial analysis in Fig 5(a), the area point pairs have an average distribution around the 1:1 line, where half the points are close to the 1:1 line. The slope of the regression equation is 1.02, yielding a coefficient of determination of 0.96. From Fig 5(b), according to the analysis of the city, most area point pairs also have a distribution around the 1:1 line. However, compared with the provincial statistical results, the accuracy in each city is lower. The slope of the regression equation is 1.17, yielding a coefficient of determination of 0.83.

Table 3 lists the statistics of the consistency indices for all years in each city. The results show that the classification accuracies of De Zhou, He Ze, Ji Nan, and Liao Cheng are high, such that, over the eight year period, they have consistency indices > 80%. Five years were characterized by a consistency index < 40% in Lai Wu, Dong Ying, Wei Hai, and Ri Zhao. The results show that there are numerous plain areas in the city with high classification accuracy while the cities with lower precision are mostly hilly areas and coastal saline areas. Larger areas of wheat and maize planting, result in higher classification result accuracies, whereas smaller planting areas yield more dispersion and lower accuracy classification results. Overall, the areas with lower precision were not the main wheat and maize production areas, indicating that this method can extract the winter wheat–summer maize planting area with high precision.

## Spatiotemporal change analysis of winter wheat–summer maize

**Area change analysis.** Fig 6 shows the area statistics of the wheat and maize in Shandong Province over the study period. The minimum planting area of the winter wheat–summer

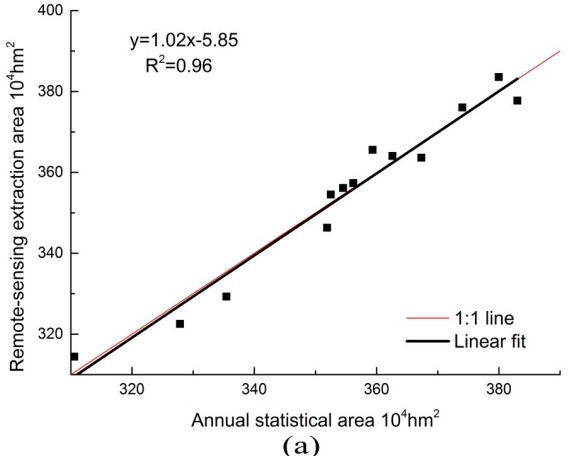
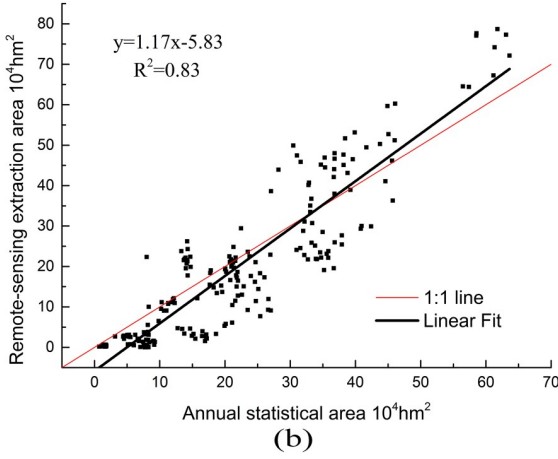

(a)　　　　(b)

**Fig 5. A comparison of the winter wheat planting area between the remote sensing monitoring data and agricultural statistics data.** (a) Statistics by province and (b) statistics by city.

**Table 3. Each city's statistical consistency for all years.**

| City | Cropland percentage | Terrain | Consistency index | | | | |
|---|---|---|---|---|---|---|---|
| | | | > 80% | 60–80% | 40–60% | 20–40% | < 20% |
| De Zhou | 14.05% | Plains | 9 | 3 | 1 | 0 | 0 |
| He Ze | 13.69% | Plains | 7 | 6 | 0 | 0 | 0 |
| Liao Cheng | 12.20% | Plains | 8 | 5 | 0 | 0 | 0 |
| Wei Fang | 9.50% | Plains | 1 | 7 | 3 | 2 | 0 |
| Ji Ning | 9.37% | Plains | 4 | 5 | 4 | 0 | 0 |
| Bin Zhou | 9.19% | Plains | 1 | 3 | 3 | 1 | 5 |
| Ji Nan | 6.03% | Plains and hills | 9 | 3 | 1 | 0 | 0 |
| Tai'an | 4.66% | Plains and hills | 8 | 5 | 0 | 0 | 0 |
| Lin Yi | 3.68% | Plains and hills | 4 | 6 | 2 | 1 | 0 |
| Qing Dao | 3.61% | Plains and hills | 1 | 4 | 4 | 2 | 2 |
| Zibo | 3.42% | Plains and hills | 8 | 4 | 1 | 0 | 0 |
| Dong Ying | 2.99% | Plains | 1 | 1 | 5 | 6 | 0 |
| Yan Tai | 2.29% | Hills | 0 | 1 | 3 | 2 | 7 |
| Zao Zhuang | 2.71% | Plains and hills | 0 | 6 | 6 | 0 | 1 |
| Wei Hai | 1.04% | Hills | 0 | 0 | 2 | 2 | 9 |
| Ri Zhao | 0.83% | Hills | 0 | 1 | 2 | 3 | 7 |
| Lai Wu | 0.74% | Hills | 0 | 1 | 1 | 3 | 7 |

maize was 3.14 Mha (2004) while the maximum was 3.83 Mha (2015), with an average area of 3.55 Mha. From 2004 to 2016, the planting area of winter wheat and summer maize in Shandong Province showed a sustained growth trend. The planting area of wheat and maize increased the fastest from 2006 to 2008, with an average growth rate of 4.93%, followed by an average growth rate of 2.7% from 2014 to 2015. From 2009 to 2013, the growth of the planting area slowed, with an average growth rate of only 0.5%. Compared with 2004, the wheat and maize planting area increased by 0.69 Mha in 2016, with an average growth rate of 1.99%.

**Spatiotemporal patterns in the winter wheat–summer maize.** Fig 7 shows a distribution map of the winter wheat–summer maize planting in Shandong Province for 2016. Fig 7 shows that the spatial distribution of the winter wheat–summer maize planting area is quite different. Wheat and maize were mainly distributed in the plains areas of Shandong Province. The Northwest Plain of the Shandong Province, such as in De Zhou, Liao Cheng, and He Ze, was the main winter wheat–summer maize planting area. In this region, the planted area in 2016 accounted for 62.61% of the province, such that the wheat planting area was large and continuous, which was suitable for large-scale centralized planting production. Second, the north-eastern plains of the middle mountainous area, such as in Ji Nan, northern Zi Bo, southern Bin zhou, and northeast of Wei Fang, also had a large area for wheat and maize planting, accounting for 29.18% of the area. In addition, in the southern portion of the middle mountainous area at the junction of Shandong and Jiangsu Provinces, several wheat and maize areas were also distributed, accounting for 4.75% of the area. The eastern hilly regions and middle mountainous area are dominated by hills and mountains, with plains basins interlaced among them. The distribution of wheat and maize in this area was small and scattered.

**Spatiotemporal dynamic analysis.** Figs 7 and 8 show time series of the winter wheat–summer maize spatial distribution from 2004 to 2016. The spatial pattern of the main period of wheat and maize planting from 2004 to 2016 was generally stable. Concentrated planting areas were stable in both the northwest plain of Shandong and the southern and north-eastern parts of the middle mountainous area, which is consistent with the crop planting topographical

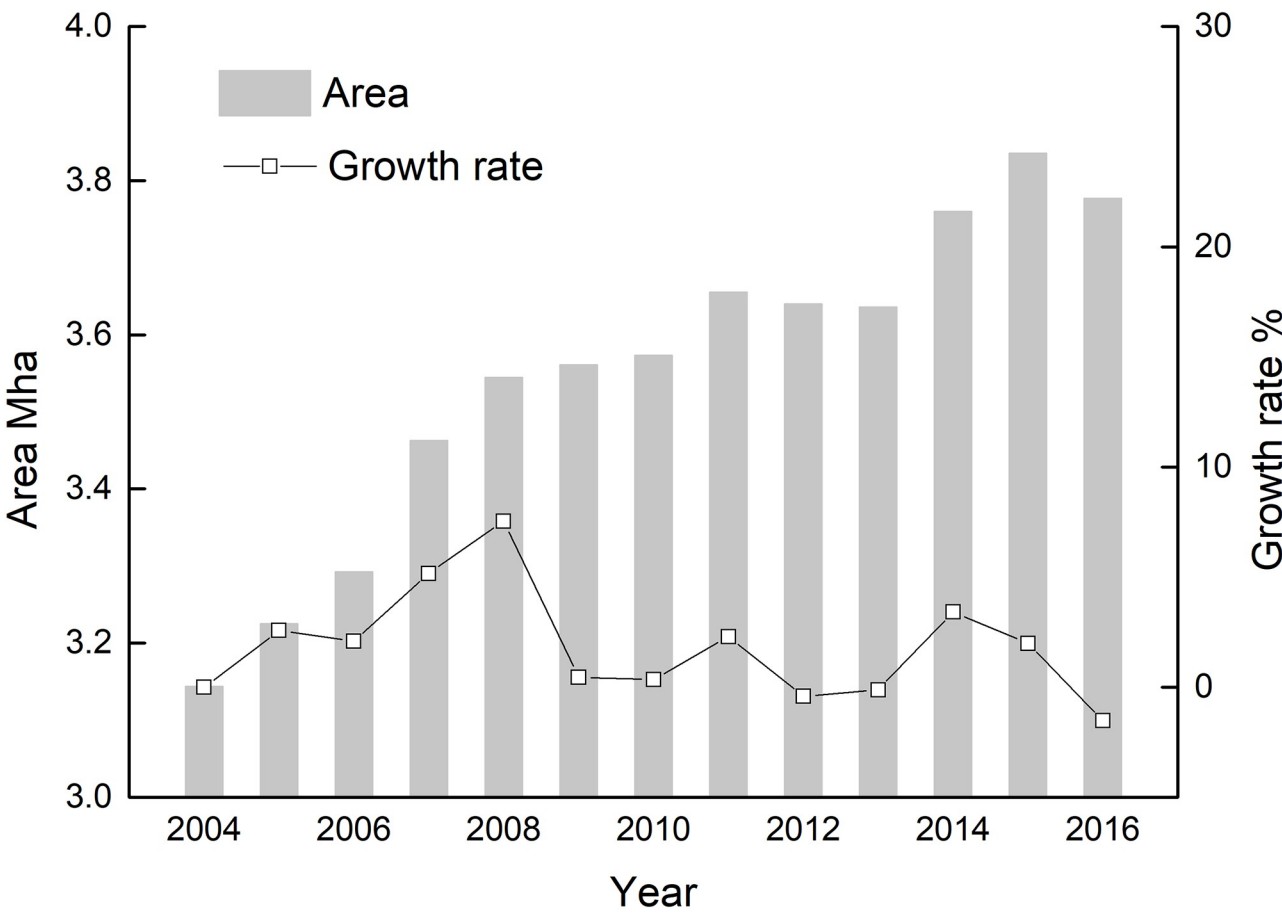

**Fig 6. The area and growth rate of the winter wheat–summer maize in Shandong from 2004 to 2016.**

constraints in Shandong Province. In addition, other winter wheat–summer maize planting areas were smaller, causing more fragmented plots. Fig 9 shows the spatial distribution of wheat and maize in the three selected periods. During the period from 2004–2008, there was a relative balance in the expansion of winter wheat–summer maize area. The north-western plains, southern region, and eastern area of the middle mountainous area had clear increases in their plantings, while areas had a decrease in their planting areas are scattered. From 2008–2012, the growth area for the winter wheat–summer maize was mainly located in the western plains of Shandong, especially in the northern region, while there were relatively noticeable decreases in the southern part of the middle mountainous areas. From 2012–2016, the winter wheat–summer maize area growth trend slowed, such that area expansion became concentrated in the northern part of De Zhou, the Yellow River Delta, and other regions. The areas with decreasing trends were scattered. Overall, the increase in the winter wheat–summer maize area mainly included the northern Shandong plain and Yellow River Delta region, while the areas that decreased had a relatively small, scattered distribution.

**Winter wheat–summer maize planting probability.** For the selection of crops affected by soil conditions, economics, government policies, and other factors, replacement sowing is commonly used during production. Crop planting on specific plots can have large interannual uncertainties. Fig 10 shows the number and area of winter wheat–summer maize planting throughout the 13-year period. The proportion of areas that planted winter wheat–summer

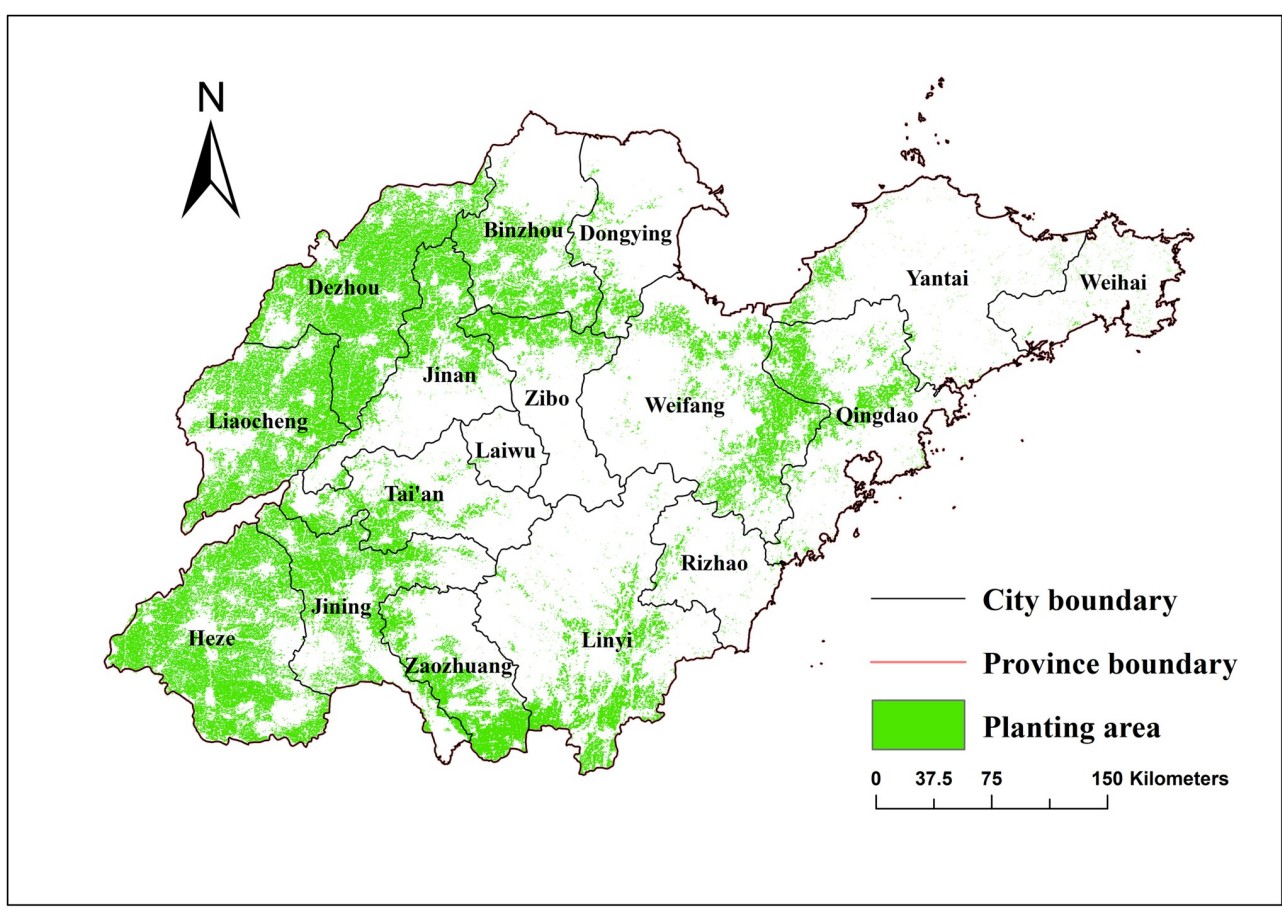

**Fig 7. Distribution map of the winter wheat–summer maize planting in 2016.**

maize with a 100% frequency was much larger than that of other planting frequencies. More than 70% of the total study area planted winter wheat–summer maize with a greater than 80% frequency. Fig 11 shows the spatial distribution of different planting probabilities for the winter wheat–summer maize. The area that had a planting frequency greater than 85% was mainly distributed in the western plains and plains surrounding the middle mountainous area, i.e., a distribution area that was large and contiguous. Considering that the winter wheat–summer maize planting area increased by 23.3% over the past 13 years, the actual planting probability in the region is even higher. These regions with high planting frequencies are the core cropland areas in Shandong Province, which provides a stable guarantee for grain production. The areas with low planting probability are mainly located around cities and hilly areas, as well as in regions with recent increases in their areas. These are limited by urban expansion, transportation costs, and high cultivated land fragmentation.

**Analysis of NDVI variations in wheat and maize.** Fig 12(a) shows the NDVI time series curve and its peak winter wheat–summer maize changes from 2004–2016. The figure shows that the NDVI curve of winter wheat during its growth period is generally scattered. The node of the growth period changed significantly in different years, such that the maximum NDVI value and peak time were also different. The peak winter wheat range was between the days 108 and 123 in the year, i.e., from mid–late March to early April. The time series curve of the maize was relatively concentrated, such that the growth period and maximum NDVI value for

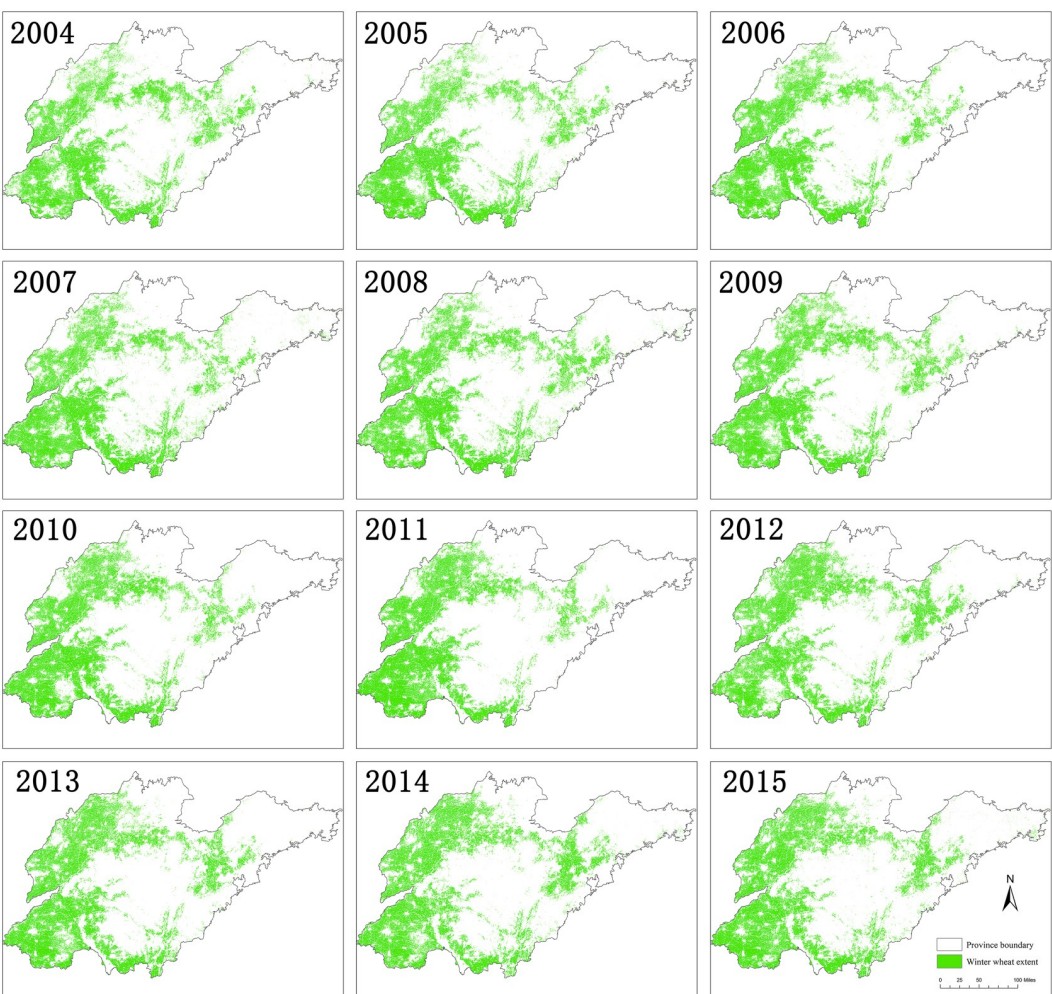

**Fig 8. The spatial distribution of the winter wheat–summer maize in Shandong from 2004 to 2015.**

different years were relatively close. The peak area of the maize NDVI was between the days 221 and 228 of the year (i.e., mid-July), which indicates that environmental factors significantly affect the growth of wheat in the winter and spring while the growth of maize in the summer and autumn has less of an environmental impact. Fig 12(b) shows the NDVI time

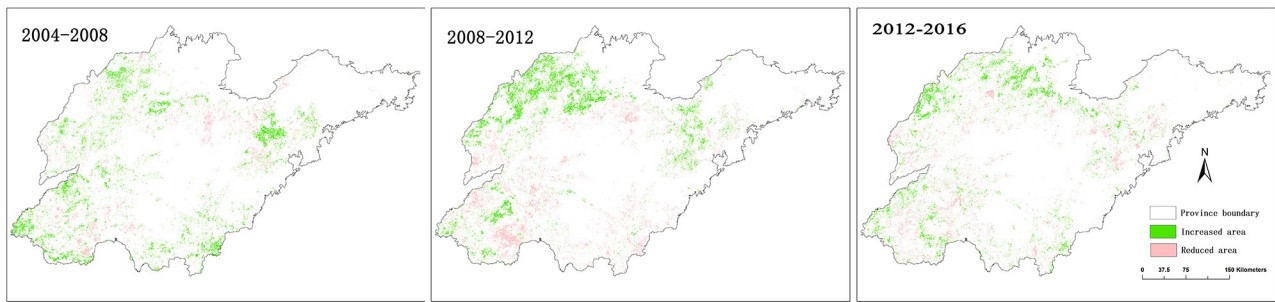

**Fig 9. Changes in the spatial distribution of the winter wheat–summer maize during different time periods.**

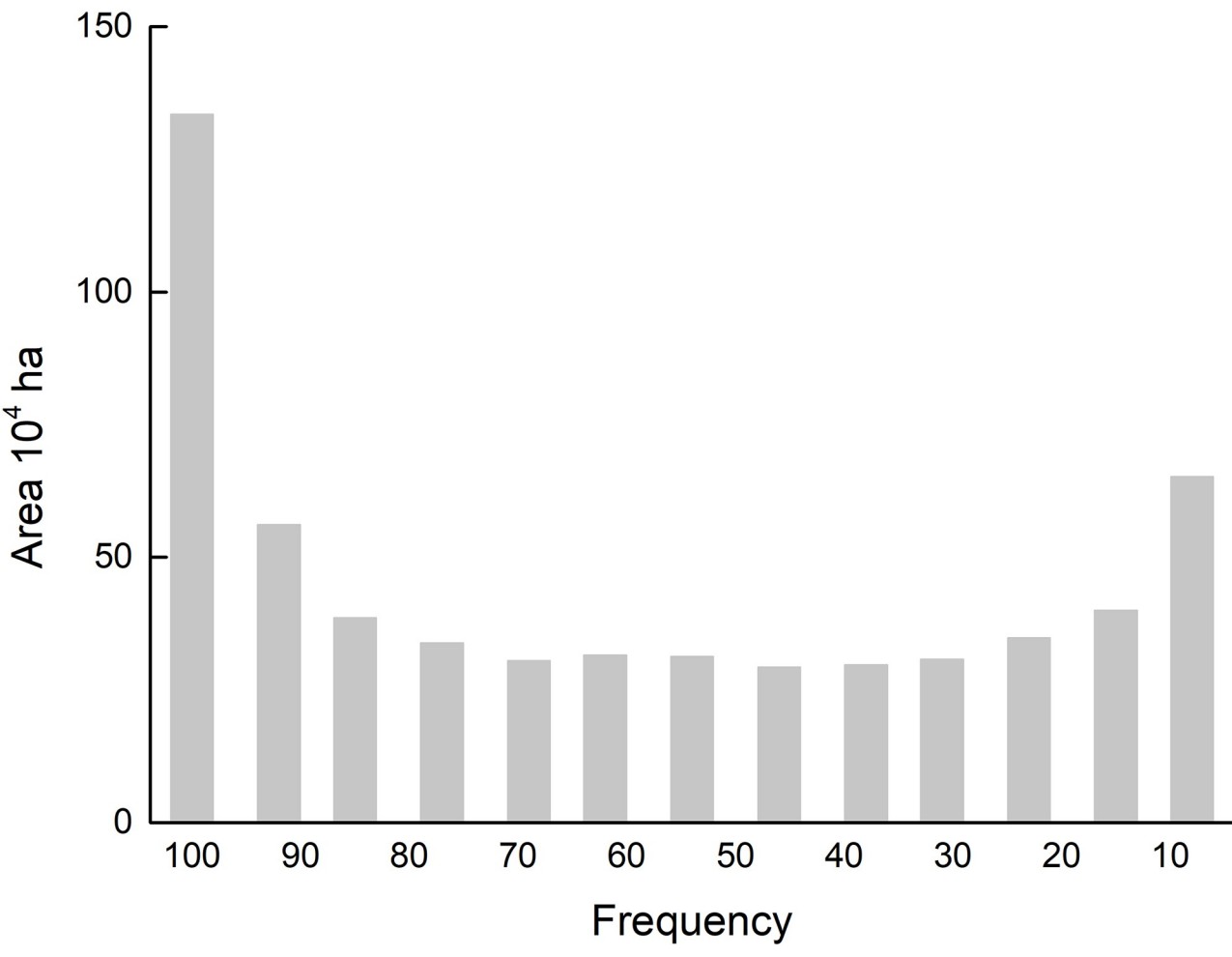

**Fig 10. Frequency area statistics in Shandong from 2004 to 2016.**

series curve profiles of winter wheat and summer maize. The figure shows that the peak NDVI of winter wheat increased significantly from 2004 to 2008 and continued to rise after a decline in 2009, showing an overall increasing NDVI peak trend. The NDVI peak value of the maize was relatively stable, such that the overall change was not significant, showing a steady and slightly rising trend. This may suggest that the production level of the winter wheat in Shandong Province has increased, while that of the summer maize has remained relatively stable during the study period.

## Discussion

This study extracted spatiotemporal distribution information from large-scale winter wheat–summer maize planting from the MODIS-NDVI time series dataset. The results showed that the extracted planting area of winter wheat and summer maize was in good agreement with field and consistent with results from previous studies that used remote sensing data or land statistical data [37]. The results show that the phenological characteristic of the winter wheat–summer maize was significantly different from that of other land cover, which can be used as a significant indicator in crop remote sensing monitoring. Considering the demand for large-

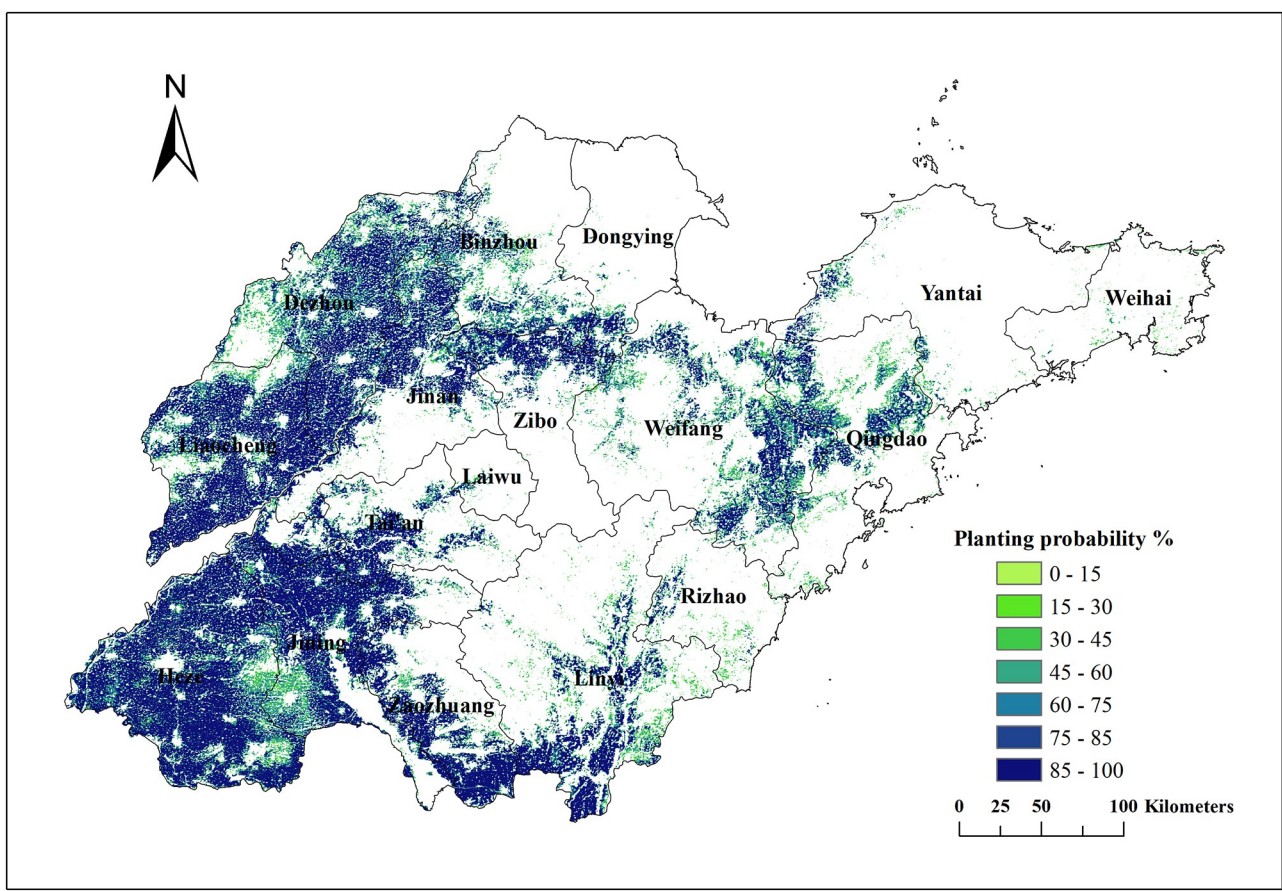

**Fig 11. Winter wheat–summer maize planting probability in Shandong from 2004 to 2016.**

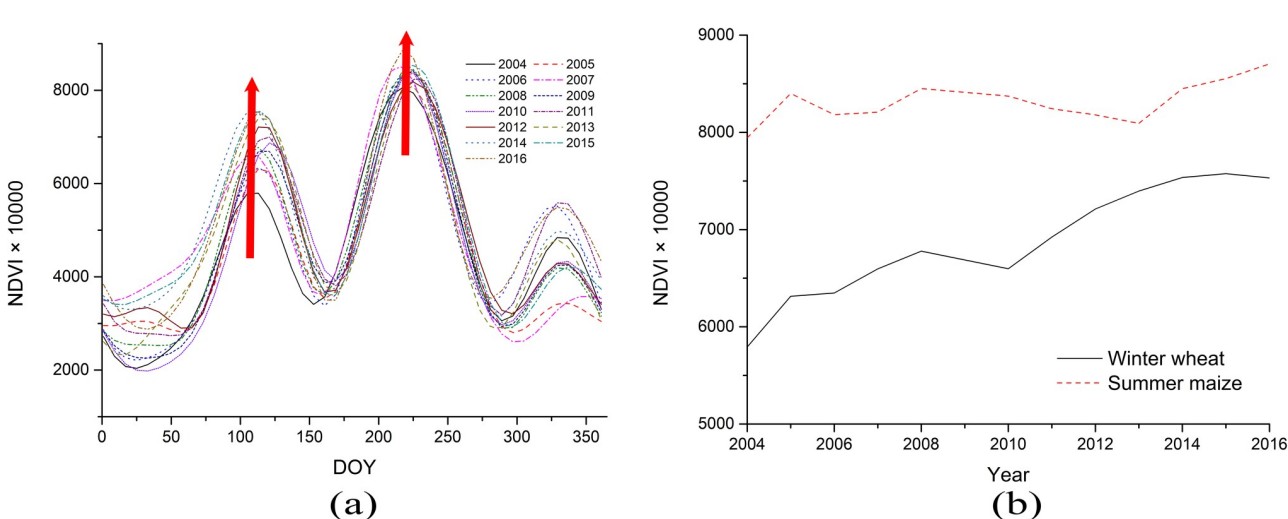

**Fig 12. Time series (a) and the curve profile (b) of winter wheat and summer maize from 2004 to 2016.** (DOY: day of year).

scale wheat–maize double-cropping rotation and other crop planting management in northern China, this method can be applied to a wider area of northern China, such that we can further subdivide other crops with unique phenological indicators, such as cotton, rice, and sweet potatoes. With the support of high-resolution remote sensing and ground data, we are able to obtain more accurate results [38, 39].

Three main factors affect the accuracy of winter wheat–summer maize area extractions from the time series within the remote sensing data. The first factor is the construction of high-quality time series datasets. In this study, the use of data from two satellites can guarantee more time nodes in a phenological period, which yields improved fault tolerance for time series data construction. At the same time, the image is pre-processed with quality data to avoid influences from abnormal values, which are marked as aberration metadata, thereby significantly improving the quality of the data smoothing. Second, the data smoothing (HANTS) method can reduce spectral complexity caused by different agroclimatic conditions and crop phenology, highlight phenological characteristics, and simplify curve clustering. At the same time, the time series data contains not only the spatial distribution characteristics of crops, but also the temporal characteristic, which is equivalent to adding a significant phenological indicator in the extraction. More information means higher classification accuracy. With the accumulation of remote sensing data, temporal features should be taken as the basic important parameters to participate in remote sensing crop monitoring. Third, the use of efficient classifiers improved the data accuracy. RF has high prediction accuracy, good tolerance to outliers and noise, a faster training speed and an improved classification effect as compared with other algorithms [40].

Based on an analysis of the area classification accuracy, we found that the crop area advance accuracy of the MODIS-NDVI data is related to the study scale, such that the accuracy of the provincial scale was significantly higher than that of the prefecture and city scale. In addition, for the relatively complex topography of the eastern hilly and middle mountainous areas, the land use heterogeneity was higher, resulting in more mixed pixels and lower classification reliability. Therefore, this data method is suitable for large-scale areas above the provincial level. To improve the classification accuracy, future studies can be done from a mixed-pixel decomposition perspective [41].

This study has shown that, from 2004 to 2016, the planting area of winter wheat–summer maize in Shandong Province continuously increased while gradually and decreasing. The northern part of the western plains and Yellow River Delta were the main areas of growth. Improvements to saline–alkali land in the region should be one of the important driving forces of this increase in area. The NDVI index value of the wheat and maize growth period indicated an upward trend, whereas the cultivated land area in Shandong Province decreased during this period [42]. This reflects the importance of grain production in the region. In the future, Shandong Province should scientifically perform grain production planning, further stabilize the western plains where winter wheat–summer maize are spatially concentrated at large scales with high planting probability, improve the agricultural intensity level, promote new technologies such as precision fertilization and water-saving irrigation, maintain the quality of the cultivated land, and protect the agricultural ecological environment.

## Conclusions

In this study, we used time series remote sensing data to extract the winter wheat–summer maize planting area in Shandong Province using a random forest classification method and analyzed the spatiotemporal changes in the planting areas from 2004–2016. The results of our study provide the following conclusions.

Using a combination of MODIS data from two satellites to construct a time series dataset and linearly interpolate quality data with the HANTS smoothing method, we can remove the abnormal fluctuation of pixel values caused by a satellite sensor's own performance or clouds, snow, and atmospheric conditions, such that the smoothing curve can highlight the phenological characteristics of the winter wheat–summer maize.

A method based on the MODIS-NDVI time series data was developed to extract the winter wheat–summer maize with the random forest classification based on the phenological characteristics of the surface cover. The average user's accuracy when using this method was 94.5% while the producer's accuracy was 90.2%. The area determination coefficient from the remote sensing extraction and city statistical yearbooks was 0.96, such that the area distribution was consistent with reality. This method can provide a pattern for remote sensing monitoring of grain crops in northern China.

The spatial distribution of winter wheat–summer maize in Shandong Province was generally stable and relatively concentrated, with predominant distribution in the western plains and north-eastern and southern plains of the middle mountainous areas. The average planting area of winter wheat–summer maize was 3.55 Mha, which increased steadily and continuously. The most noticeable growth areas were mainly located in the northern area of the western plains and Yellow River Delta. The planting probability map better reflects the core area of winter wheat–summer maize in Shandong Province. During the study period, the peak time of winter wheat significantly varied. The peak NDVI value for summer maize was relatively stable while the peak NDVI value for winter wheat showed an upward trend.

In this study, we proposed an effective method of wheat- and maize-area extraction based on NDVI time series data. We were able to sufficiently define the area of winter wheat and summer maize in Shandong Province, as well as the temporal and spatial changes over the past 13 years, which provides a scientific basis for wheat and maize planting management. Future studies, must further improve the accuracy of the winter wheat–summer maize area by enhancing mixed-pixel recognition, obtaining more ground data verification, and introducing more quantitative models to analyze and predict spatiotemporal distribution patterns.

## Acknowledgments

The MODIS/Terra Vegetation Indices 16-Day L3 Global 250m SIN Grid and MODIS/Aqua Vegetation Indices 16-Day L3 Global 250m SIN Grid datasets were acquired from the Level-1 and Atmosphere Archive &Distribution System (LAADS) Distributed Active Archive Center (DAAC), located at the Goddard Space Flight Center in Greenbelt, Maryland(https://ladsweb.nascom.nasa.gov/). We would like to thank Editage (www.editage.cn) for English language editing and the anonymous reviewers for reviewing this manuscript and providing comments.

## Author Contributions

**Conceptualization:** Gengxing Zhao.

**Data curation:** Chao Dong.

**Formal analysis:** Chao Dong.

**Funding acquisition:** Gengxing Zhao.

**Investigation:** Chao Dong.

**Software:** Hong Wan.

**Supervision:** Gengxing Zhao.

**Visualization:** Hong Wan.

**Writing – original draft:** Chao Dong.

**Writing – review & editing:** Gengxing Zhao, Yuanwei Qin.

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
