## [Decision Letter · Decision Letter 0]

27 Sep 2019

PONE-D-19-20112

Area extraction and spatiotemporal characteristics of winter wheat–summer maize in Shandong Province using NDVI time series

PLOS ONE

Dear Dr Zhao,

Thank you for submitting your manuscript to PLOS ONE. After careful consideration, we feel that it has merit but does not fully meet PLOS ONE’s publication criteria as it currently stands. Therefore, we invite you to submit a revised version of the manuscript that addresses the points raised during the review process.

The structure of the introduction need to be revised according to reviewers comments to put the study in the proper context. The description provided in each of the sections also need to be reviewed to make sure that only important information is retained. Description of important methodological parameters and their justification is missing for example. The discussion is also currently weak and needs to be strengthened by explaining the important aspects of the presented approach for practical application. 

We would appreciate receiving your revised manuscript by Nov 11 2019 11:59PM. To enhance the reproducibility of your results, we recommend that if applicable you deposit your laboratory protocols in protocols.io, where a protocol can be assigned its own identifier (DOI) such that it can be cited independently in the future. For instructions see: http://journals.plos.org/plosone/s/submission-guidelines#loc-laboratory-protocols

We look forward to receiving your revised manuscript.

Kind regards,

Abel Chemura

Academic Editor

PLOS ONE

Journal Requirements:

2. We note that Figures in your submission contain map/satellite images which may be copyrighted. All PLOS content is published under the Creative Commons Attribution License (CC BY 4.0), which means that the manuscript, images, and Supporting Information files will be freely available online, and any third party is permitted to access, download, copy, distribute, and use these materials in any way, even commercially, with proper attribution. For these reasons, we cannot publish previously copyrighted maps or satellite images created using proprietary data, such as Google software (Google Maps, Street View, and Earth). For more information, see our copyright guidelines: http://journals.plos.org/plosone/s/licenses-and-copyright.

1.    You may seek permission from the original copyright holder of Figures to publish the content specifically under the CC BY 4.0 license. 

Reviewers' comments:

Reviewer's Responses to Questions

**Comments to the Author**

1. Is the manuscript technically sound, and do the data support the conclusions?

Reviewer #1: Partly

Reviewer #2: Yes

2. Has the statistical analysis been performed appropriately and rigorously? 

Reviewer #1: Yes

Reviewer #2: Yes

3. Have the authors made all data underlying the findings in their manuscript fully available?

Reviewer #1: Yes

Reviewer #2: Yes

4. Is the manuscript presented in an intelligible fashion and written in standard English?

Reviewer #1: Yes

Reviewer #2: Yes

5. Review Comments to the Author

Reviewer #1: Authors of the manuscript entitled “Area extraction and spatiotemporal characteristics of winter wheat–summer maize in Shandong Province using NDVI time series” have presented a feasible approach to the mapping of crop rotation patterns involving wheat and maize using NDVI time series obtained from MODIS. However, the paper needs major improvements as given below:

1. The methodological novelty is not well articulated in the abstract and introduction.

2. The description of the study area can be reduced.

3. Paragraphs in the conclusion section should not be numbered.

4. Define the mtry and ntry parameters used in running the RF model

5. Proper English language editing is also required.

Reviewer #2: General

The importance of food crops such as maize and wheat is well known, and there is no need for motivation on that, what is lacking in the introduction is the motivation why is it so important to study the phenology of these food crops.

In the introduction, the authors introduce MODIS as a medium spatial resolution sensor (Page 3 line 54) and a few moments later, MODIS is now a low spatial resolution sensor (Page 3 line 58), please rectify. Also explain how do phenology metrics solve the problem of low spatial resolution?

The authors claimed to have used phenological metrics in this study. It is not clear which phenological metrics were used and why, how were these phenological metrics extracted, which algorithm (s) were used? All this information should be clear in the methods section.

Figure 4 shows NDVI temporal curves for five categories, the main objective of this study was to extract spatiotemporal characteristics of winter wheat–summer maize, therefore l do not see the relevance of the other categories presented here.

The description citing figure 4 is using months but the graphical presentation on figure 4 is using days, for the sake of consistency, l advice the authors to use either days or moths for both.

In the discussion, the authors claim that they have developed a tool for extracting spatiotemporal distribution information from large-scale winter wheat–summer maize planting from the MODIS-NDVI time series. However, the ‘newly developed’ tool was not mentioned, please clarify

Specific

Page 3 line 50: Give examples of the high spatial resolution images you are referring to

Page 4 line 66-68: The works of Li et al. [15] do not add value to the current work if their research findings are not reported, mentioning what their research focus only is not good enough

Page 8: Under the ‘smooth processing of time series’ section please provide a graphical presentation of raw and smoothed NDVI temporal data

Page 11 line 214: ‘land types’ do you mean land cover types or land use types?

Page 20 line 367: Replace ‘good agreement with reality’ with ‘good agreement with field/ground observation/measurements

6. PLOS authors have the option to publish the peer review history of their article (what does this mean?). If published, this will include your full peer review and any attached files.

Reviewer #1: Yes: Dr. Lamin R. Mansaray

Reviewer #2: No

---

## [Author Response · Author response to Decision Letter 0]

29 Oct 2019

Dear Editor:

Thank you for handling our manuscript entitled “Area extraction and spatiotemporal characteristics of winter wheat–summer maize in Shandong Province using NDVI time series” (No.: PONE-D-19-20112). We appreciate the comments from the reviewers, which helped to improve the manuscript significantly. In the “Response to Reviewers.docx”, we explain in detail how we responded to each of the comments.

---

## [Decision Letter · Decision Letter 1]

20 Nov 2019

PONE-D-19-20112R1

Area extraction and spatiotemporal characteristics of winter wheat–summer maize in Shandong Province using NDVI time series

PLOS ONE

Dear Dr Zhao,

Thank you for submitting your manuscript to PLOS ONE. After careful consideration, we feel that it has merit but does not fully meet PLOS ONE’s publication criteria as it currently stands. Therefore, we invite you to submit a revised version of the manuscript that addresses the points raised during the review process.

We would appreciate receiving your revised manuscript by Jan 04 2020 11:59PM. To enhance the reproducibility of your results, we recommend that if applicable you deposit your laboratory protocols in protocols.io, where a protocol can be assigned its own identifier (DOI) such that it can be cited independently in the future. For instructions see: http://journals.plos.org/plosone/s/submission-guidelines#loc-laboratory-protocols

We look forward to receiving your revised manuscript.

Kind regards,

Abel Chemura

Academic Editor

PLOS ONE

Reviewers' comments:

Reviewer's Responses to Questions

**Comments to the Author**

1. If the authors have adequately addressed your comments raised in a previous round of review and you feel that this manuscript is now acceptable for publication, you may indicate that here to bypass the “Comments to the Author” section, enter your conflict of interest statement in the “Confidential to Editor” section, and submit your "Accept" recommendation.

Reviewer #1: All comments have been addressed

Reviewer #2: All comments have been addressed

2. Is the manuscript technically sound, and do the data support the conclusions?

Reviewer #1: Yes

Reviewer #2: Yes

3. Has the statistical analysis been performed appropriately and rigorously? 

Reviewer #1: Yes

Reviewer #2: Yes

4. Have the authors made all data underlying the findings in their manuscript fully available?

Reviewer #1: Yes

Reviewer #2: Yes

5. Is the manuscript presented in an intelligible fashion and written in standard English?

Reviewer #1: Yes

Reviewer #2: Yes

6. Review Comments to the Author

Reviewer #1: Authors have fully responded to my comments and I therefore recommend the current version for publication, subject to the recommendations of other reviewers.

Reviewer #2: Figure 4 does not have a legend, what is the red dotted and the black line representing?

Page 10 line 191: I suggest you change the section ‘Accuracy validation’ to accuracy assessment

Page 10 line 194: Consider rephrasing the sentence which begins with ‘To evaluate the comprehensiveness and objectivity of..’ since the previous sentence begins exactly the same

Page 10 line 196-197: ‘high-resolution images from google earth’ Which resolution are you referring to? Spatial, spectral, temporal?

Page 11 line 216: The first heading on the results section needs attention, which remote sensing ‘approaches’ are you referring to? The authors should specify the remote sensing approaches they used

7. PLOS authors have the option to publish the peer review history of their article (what does this mean?). If published, this will include your full peer review and any attached files.

Reviewer #1: Yes: Dr. Lamin R. Mansaray

Reviewer #2: Yes: Trylee Nyasha Matongera

---

## [Author Response · Author response to Decision Letter 1]

26 Nov 2019

All replies are in the "Response to Reviewers.docx" file.

---

## [Editor Report · Decision Letter 2]

2 Dec 2019

Area extraction and spatiotemporal characteristics of winter wheat–summer maize in Shandong Province using NDVI time series

PONE-D-19-20112R2

Dear Dr. Zhao,

We are pleased to inform you that your manuscript has been judged scientifically suitable for publication and will be formally accepted for publication once it complies with all outstanding technical requirements.

With kind regards,

Abel Chemura

Academic Editor

PLOS ONE
---

## [Editor Report · Acceptance letter]

5 Dec 2019

PONE-D-19-20112R2 

Area extraction and spatiotemporal characteristics of winter wheat–summer maize in Shandong Province using NDVI time series 

Dear Dr. Zhao:

I am pleased to inform you that your manuscript has been deemed suitable for publication in PLOS ONE. Congratulations! Your manuscript is now with our production department. 

With kind regards,

on behalf of

Dr. Abel Chemura 

Academic Editor

PLOS ONE